# Unsupervised discovery of family specific vocal usage in the Mongolian gerbil

**Ralph E Peterson[1,2]\*, Aman Choudhri[3], Catalin Mitelut[1], Aramis Tanelus[1,2], Athena Capo-Battaglia[1], Alex H Williams[1,2], David M Schneider[1], Dan H Sanes[1,4,5,6]**

[1]Center for Neural Science, New York University, New York, United States; [2]Center for Computational Neuroscience, Flatiron Institute, New York, United States; [3]Columbia University, New York, New York, United States; [4]Department of Psychology, New York University, New York, United States; [5]Neuroscience Institute, New York University School of Medicine, New York, United States; [6]Department of Biology, New York University, New York, United States

## eLife Assessment

This **valuable** study provides an experimental paradigm and state-of-the-art analysis method for studying the existence of call types and transition differences among Mongolian gerbil families in a naturalistic environment. The analyses are **convincing**, with a thorough treatment of the acoustic data and a demonstration of the robustness of the observed effect across days. The work will likely be of interest to the auditory neuroscience and neuroethology communities.

\*For correspondence:
ralph.emilio.peterson@gmail.com

**Abstract** In nature, animal vocalizations can provide crucial information about identity, including kinship and hierarchy. However, lab-based vocal behavior is typically studied during brief interactions between animals with no prior social relationship, and under environmental conditions with limited ethological relevance. Here, we address this gap by establishing long-term acoustic recordings from Mongolian gerbil families, a core social group that uses an array of sonic and ultrasonic vocalizations. Three separate gerbil families were transferred to an enlarged environment and continuous 20-day audio recordings were obtained. Using a variational autoencoder (VAE) to quantify 583,237 vocalizations, we show that gerbils exhibit a more elaborate vocal repertoire than has been previously reported and that vocal repertoire usage differs significantly by family. By performing gaussian mixture model clustering on the VAE latent space, we show that families preferentially use characteristic sets of vocal clusters and that these usage preferences remain stable over weeks. Furthermore, gerbils displayed family-specific transitions between vocal clusters. Since gerbils live naturally as extended families in complex underground burrows that are adjacent to other families, these results suggest the presence of a vocal dialect which could be exploited by animals to represent kinship. These findings position the Mongolian gerbil as a compelling animal model to study the neural basis of vocal communication and demonstrates the potential for using unsupervised machine learning with uninterrupted acoustic recordings to gain insights into naturalistic animal behavior.

## Introduction

The field of ethology contains rich descriptions of complex behavioral actions, including a wealth of species-specific vocal repertoires. However, natural observations are often incomplete due to limitations in physical access for experimenter observation or behavioral recording. This can be particularly severe for family behaviors which occur in protected or remote environments, such as burrows in the case of fossorial rodent species like naked mole-rats and Mongolian gerbils (*Brett, 1986*; *Scheibler*

**eLife digest** Every time you speak, the sounds coming out of your mouth may carry more meaning that you may have intended; they may reveal, for example, which country, city or even neighborhood you may be coming from. Indeed, the vocal patterns that humans use to communicate differ from one population to the next, creating an array of languages, dialects and accents.

Such diversity has also been identified in various social species across the animal kingdom. Naked mole rats, for instance, which live underground in complex societies, exhibit different 'dialects' depending on their group of origin. Yet studying the vocal patterns of animals has remained difficult, especially for species inhabiting burrows or other environments difficult to access.

Aiming to bypass these limitations, Peterson et al. adopted a 'naturalistic' approach that allowed them to capture the vocal calls of three families of Mongolian gerbils living undisturbed in enclosures that mimic features of their natural environment. These animals spend their lives underground in tight-knit families, with multiple groups often being in close proximity. Researchers have speculated that individuals may rely on vocal cues to identify whether they are part of the same colony, as they are often too far from each other to rely on sight or smell.

Over half a million vocalizations obtained continuously through the course of 20 days were analyzed using an artificial intelligence technique known as unsupervised machine learning. The analyses helped add new types of calls to the gerbil vocal repertoire, but also highlighted its complexity. In particular, they revealed that the animals could combine individual vocal elements into complex sequences. More importantly, this approach showed that gerbil families have vocal dialects that are stable across weeks, with each group displaying a preference for certain call types (i.e. words) and certain sequential patterns (i.e. phrases).

These findings demonstrate the benefits of the approach developed by Peterson et al. for the study of animal vocalizations. Going forward, they also suggest that the Mongolian gerbil could be used as an animal model to study the neural basis of vocal communication.

---

*et al., 2006*). Some of these limitations have been addressed with laboratory environments that partially recapitulate real-world features (*Shemesh and Chen, 2023*). However, these studies generally focused on relatively short periods of data collection that consider single animals or dyads with no prior social relationship.

While our understanding of social aural communication is sparse, even for humans (*Pagel et al., 2013*; *Mascaro et al., 2018*; *Schindler et al., 2022*), we know that many vocal cues are learned through social experience, and provide pivotal information about an animal's identity. For example, a human infant's ability to discriminate between foreign language phonemes can be preserved by exposure to a live foreign speaker, but not an audiovisual recording (*Kuhl et al., 2003*). Evidence from swamp sparrows suggests the presence of culturally transmissible 'dialects' – a term borrowed from linguistics to denote a pattern of vocal behavior that is used by members of a social group (*Maler and Tamura, 1964*). Our study adopts this operational definition of a vocal dialect. Even some rodents, such as the naked mole rat, learn colony-specific dialects based on early social experience (*Barker et al., 2021*). The literature for social facilitation of vocal discrimination or production is particularly strong for zebra finches (*Eales, 1989*; *Derégnaucourt et al., 2013*; *Chen et al., 2016*; *Narula et al., 2018*). Therefore, our study considers the possibility that there is a diversity of vocalizations within the gerbil social group that may harbor family specific information.

We chose to focus on families, a canonical social group that has been predominantly studied during brief and experimentally restricted social encounters (e.g. mating, pup retrieval, aggression) in relatively featureless environments. Our goal was to construct a complete gerbil family social-vocal soundscape during a significant period of development under undisturbed, environmentally enriched conditions. Unlike many laboratory rodents, gerbils form pair bonds and maintain a family structure across generations (*Ågren, 1984a*). These families are composed of a founding adult pair, and up to 15 extended family members that live cooperatively in underground burrows (*Ågren et al., 1989a*; *Ågren et al., 1989b*; *Milne-Edwards, 1867*; *Scheibler et al., 2004*). Given the darkness and complexity of their burrow systems, gerbils are thought to rely heavily on their auditory system for social interactions. Sibling bonds established through adolescence facilitate social structure and minimize inbreeding

(*Ågren, 1984b*). Natural burrows are found in multi-family neighborhoods with strictly enforced territorial boundaries (*Scheibler et al., 2006*; *Ågren et al., 1989a*; *Ågren et al., 1989b*). Like prairie voles, gerbils act cooperatively to hoard food, maintain nests, defend their territory, and care for pups (*Elwood, 1975*; *Gromov, 2021*). Therefore, gerbils display a range of rodent-typical behaviors (*Hurtado-Parrado et al., 2017*), as well as complex family behaviors. Gerbils also display significant vocal communication in both the ultrasonic and sonic ranges (*Rübsamen et al., 2012*; *Kobayasi and Riquimaroux, 2012*) which is likely to be integral to social behaviors. Unlike many other rodent species, gerbils are able to hear within sonic ranges at sensitivities similar to humans (*Ryan, 1976*). As a result, there is a rich, contemporary literature on the auditory perceptual skills, peripheral and central physiology, central anatomy, learning, and genomics in this species (*Budinger and Scheich, 2009*; *Buran et al., 2014*; *Happel et al., 2014*; *Myoga et al., 2014*; *Pachitariu et al., 2015*; *Sarro et al., 2015*; *von Trapp et al., 2016*; *Caras and Sanes, 2017*; *Cheng et al., 2019*; *Zorio et al., 2019*; *Yao et al., 2020*; *Amaro et al., 2021*; *Paraouty et al., 2021*; *Yao and Sanes, 2021*; *Saldeitis et al., 2022*; *Penikis and Sanes, 2023*).

Here, we made continuous 20-day audio recordings from three separate gerbil families (2 parents, 4 pups) in an enlarged home cage that was isolated from other gerbils and humans. Specifically, we recorded audio over a period beginning at postnatal day 11–13 when auditory cortex is particularly sensitive to acoustic experience, and extending to postnatal day 31–32, the time when animals are typically weaned. Our goal was to acquire a descriptive dataset of the spectrotemporal structure of vocalizations emitted throughout daily family life, and without human intervention. Using emerging methods in unsupervised vocalization analysis, we quantitatively describe the spectrotemporal structure of vocalizations over multiple timescales and demonstrate that vocal repertoire usage differs between families.

## Results

### Longitudinal familial audio recording

We obtained acoustic recordings (four microphones, 125 kHz sampling rate) from three separate gerbil families, each containing two adults and four pups (*Figure 1A*). Continuous recordings began at P11-13, lasted 20 days, and pups were weaned at P29 (*Figure 1B*). As shown in *Figure 1C*, we extracted all sound events (yellow) using amplitude thresholding of acoustic power. To isolate vocalizations (blue) from non-vocal sounds (red), we computed the spectral flatness of each sound event and classified sounds with a threshold value of <0.3 as vocalizations. A similar approach has previously been used in mice (*Castellucci et al., 2016*), and we verified that a threshold value of 0.3 minimized the number of false positives (*Figure 1—figure supplement 1*). Using this approach, 10,267,972 sound events were extracted, containing 583,237 vocalizations and 9,684,735 non-vocal sounds detected across the three families. Sound events were produced at an average rate of 6726+/-1260 times per hour (*Figure 1D*), which reveals the rate of auditory object processing (*Griffiths and Warren, 2004*) for gerbil families in an undisturbed setting. Vocalizations represent 6.99+/-3.07% of all sound events over the recording period (*Figure 1E*) and were emitted at an average rate of 405+/-103 times per hour (*Figure 1F*), although this varied with time of day (see below).

### Unsupervised discovery of the Mongolian gerbil vocal repertoire

To quantify the full array of vocalizations obtained from the three families, we trained a variational autoencoder (VAE) on vocalization spectrograms. The VAE learned a low-dimensional representation of latent acoustic features, thereby enabling analysis of such a large dataset with a larger representational capacity than standard acoustic features (*Goffinet et al., 2021*). *Figure 2A* shows a schematic of the VAE architecture used (*Goffinet et al., 2021*), where spectrograms (top; 128x128 pixels) are reduced via a deep convolutional neural network 'encoder' to a latent vector (middle; 32-dimensional). A deep convolutional neural network 'decoder' then reconstructs a spectrogram (bottom) from the 32-dimensional latent representation. The encoder/decoder networks are jointly trained to minimize the discrepancy between the original and reconstructed spectrograms (*Figure 2—figure supplement 1A, B*), resulting in a low-dimensional latent representation, or 'code', which depicts each vocalization. To cluster vocalizations into distinct categories, we trained a Gaussian Mixture Model (GMM) on VAE latent representations. Using a combination of the elbow method on held-out log likelihood and

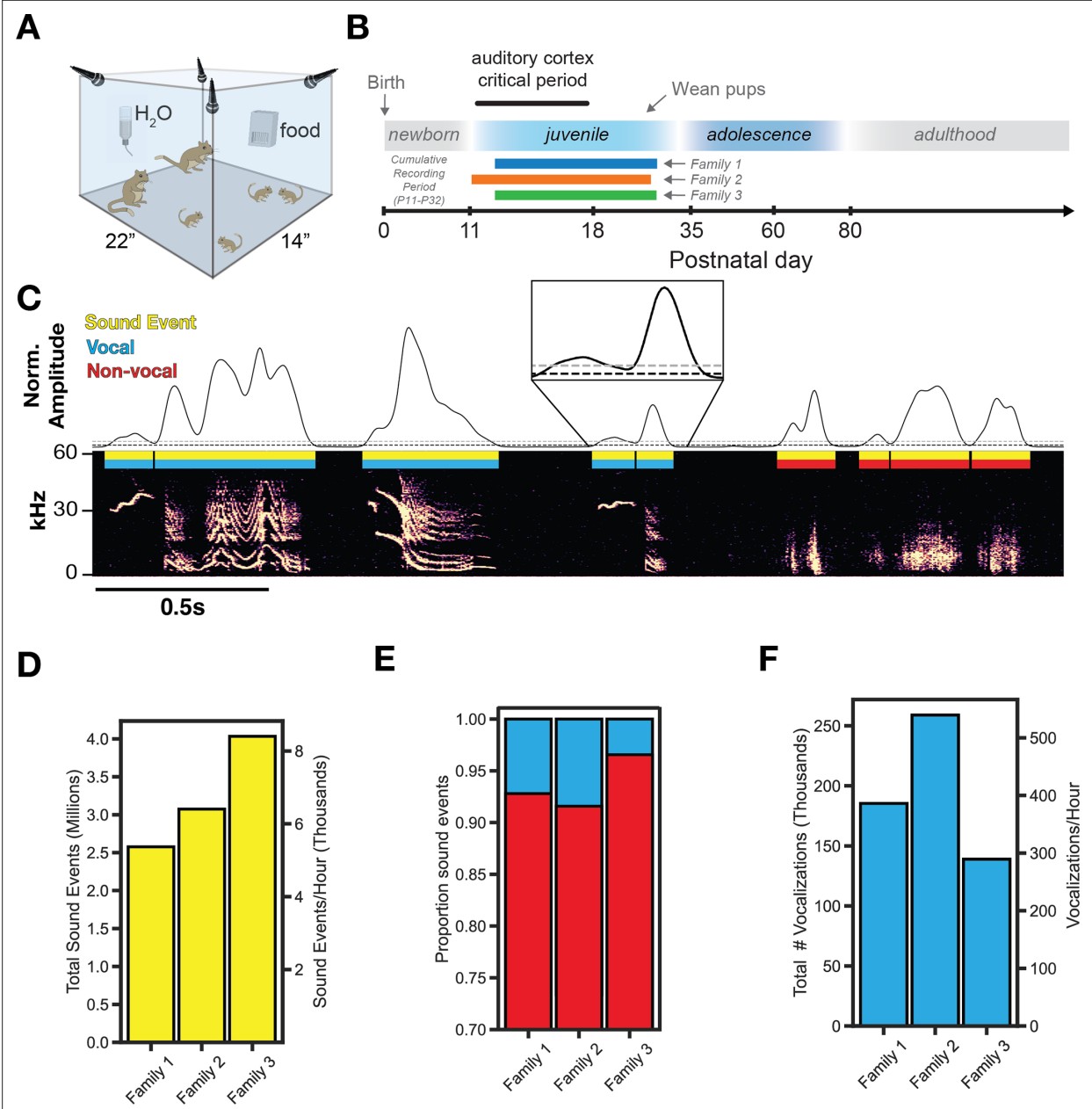

**Figure 1.** Longitudinal familial audio recording. (**A**) Recording apparatus. Four ultrasonic microphones sampled at 125 kHz continuously recorded a family in an enlarged environment. (**B**) Experiment timeline. Three gerbil families with the same family composition (2 adults, 4 pups) were recorded continuously for 20 days. (**C**) Extraction of sound events from raw audio using sound amplitude thresholding (Gray threshold = 'th_2', black threshold = 'th_1' and 'th_3'; see Methods). Vocalizations (n=583,237) are separated from non-vocal sounds (n=9,684,735) using a threshold on spectral flatness (*Figure 1—figure supplement 1* see Methods). (**D**) Summary of total sound event emission and average emission per hour. (**E**) Proportion of all sound events that are vocal or non-vocal sounds. (**F**) Summary of total vocalization emission and average emission per hour.

The online version of this article includes the following figure supplement(s) for figure 1:

**Figure supplement 1.** Vocalization extraction.

established knowledge for how many vocal types gerbils emit (see Methods, GMM clustering), we selected a model with 70 vocal clusters as a parsimonious description of the data (*Figure 2—figure supplement 1C*). *Figure 2B* shows a UMAP embedding of the VAE latents (center), used for visualization purposes only, which demonstrates that the gerbil vocal repertoire is more discrete than mouse, yet less discrete than zebra finch (*Sainburg et al., 2020*; *Goffinet et al., 2021*). Vocalizations occur as either single syllables bounded by silence (monosyllabic) or consist of combinations of single syllables

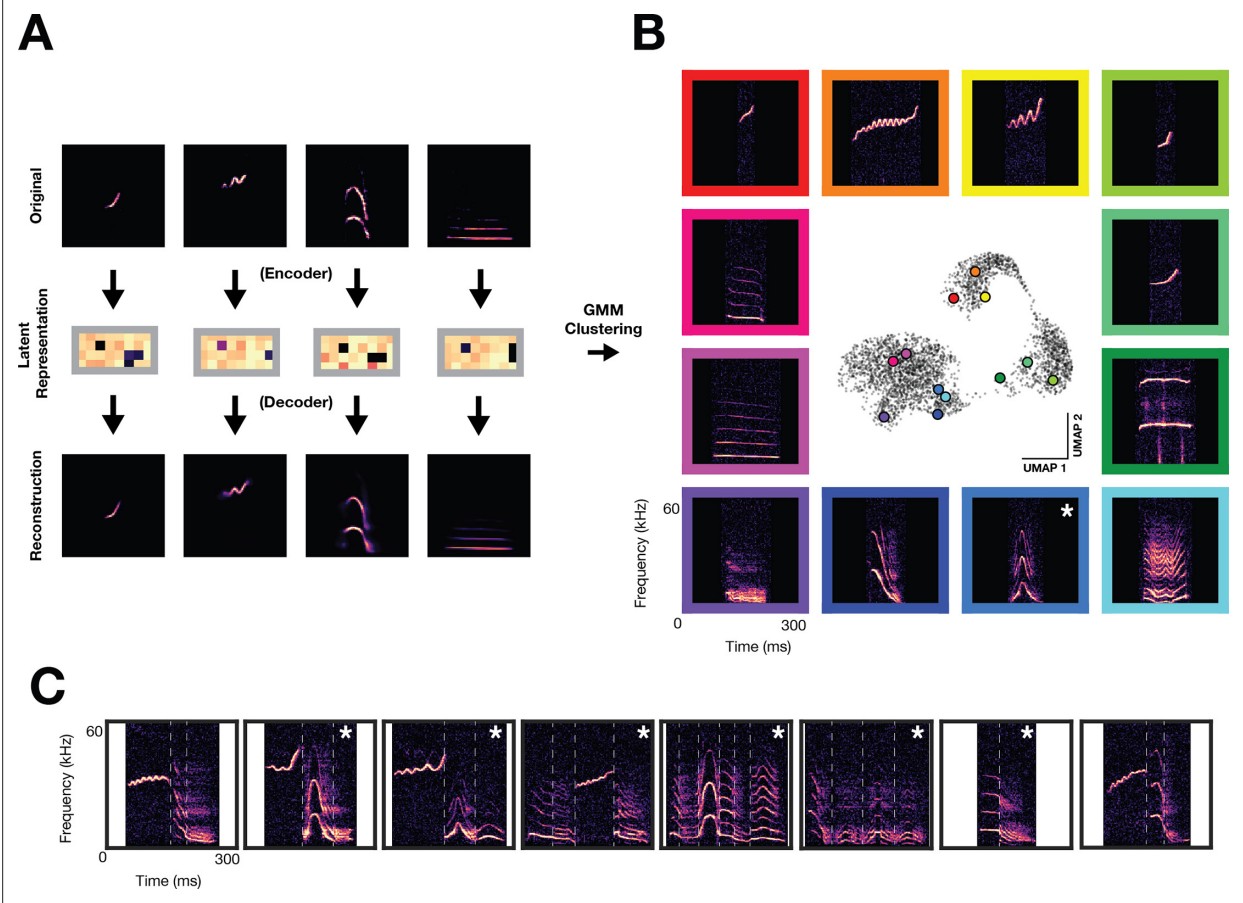

**Figure 2.** Unsupervised discovery of the Mongolian gerbil vocal repertoire. Variational autoencoder and clustering. (**A**) Vocalization spectrograms (top) are input to a variational autoencoder (VAE) which encodes the spectrogram as a 32-D set of latent features (middle). The VAE learns latent features by minimizing the difference between original spectrograms and spectrograms reconstructed from the latent features by the VAE decoder (bottom). A gaussian mixture model (GMM) was trained on the latent features to cluster vocalizations into discrete categories. (**B**) Representative vocalizations from 12 distinct GMM clusters featuring monosyllabic vocalizations are shown surrounding a UMAP embedding of the latent features. Asterisk denotes vocal type not previously characterized. (**C**) Examples of multisyllabic vocalizations. White vertical lines indicate boundaries of monosyllabic elements. Asterisks denote multisyllabic vocal types not previously characterized.

The online version of this article includes the following figure supplement(s) for figure 2:

**Figure supplement 1.** VAE training and GMM clustering.

without a silent interval (multisyllabic). Representative examples from 12 monosyllabic vocalization clusters are shown with their relative position in UMAP space, one of which appears similar in form to naked mole rat family specific chirp (blue box with asterisk; *Barker et al., 2021*). Furthermore, monosyllabic vocalizations (56/70 vocal clusters) can be flexibly strung together to create multisyllabic or 'composite' vocalizations (9/70 of vocal clusters; *Kobayasi and Riquimaroux, 2012*). The remaining five clusters contained a mixture of monosyllabic and multisyllabic vocalizations. *Figure 2C* shows 8 examples of multisyllabic vocalizations and their monosyllabic component boundaries, some of which have been reported previously (*Kobayasi and Riquimaroux, 2012*) and some of which are newly characterized (white asterisks). To assess how family structure influences vocal repertoire usage, we compared vocal usage one day prior and one day after pup weaning, showing a drastic decrease in vocal emission (*Figure 3—figure supplement 1A*). A large-magnitude vocal repertoire change is also observed, with the repertoire confined to a small region of vocal space following weaning (*Figure 3—figure supplement 1B–D*).

## Family specific usage of vocal clusters

We next asked whether gerbil families display different vocalization usage patterns. First, we visualized the entire vocal repertoire usage of each family as a probability density heatmap and determined that vocal repertoire usage significantly differed between families (*Figure 3A*, *Figure 2—figure supplement 1D*). Next, using GMM vocalization clusters, we compared the proportion usage of each vocal cluster for the three families, revealing specific vocal cluster differences between families (*Figure 3B*). All families used each of the 70 vocal types (i.e. no cluster usage is 0), but each family relied more heavily on some clusters as compared to others. Importantly, this result is stable across a wide range of GMM clusters (*Figure 4—figure supplement 1*).

Sorting the GMM cluster labels by the pairwise difference in vocal type usage between the three families revealed which vocal types differed most (*Figure 3C*). Examples of top preferred vocal clusters for each family are shown in *Figure 3D*, along with the position of those vocal clusters in UMAP embedding space. Families overexpress dissimilar vocal clusters relative to each other (e.g. clusters 4 and 8 in Family 2) and similar vocal clusters relative to each other (e.g. cluster 14 in Family 1 and cluster 1 in Family 3; cluster 9 in Family 1 and cluster 5 in Family 2).

## Vocal usage differences remain stable across days of development

It is possible that the observed vocal usage differences could result from varying developmental progression of vocal behavior or overexpression of certain vocal clusters during specific periods within the recording. To assess the potential effect of daily variation on family specific vocal usage, we visualized density maps of vocal usage across days for each of the families (*Figure 4A*). There are two noteworthy trends: (1) the density map remains coarsely stable across days (rows) and (2) the maps look distinct across families on any given day (columns). This is a qualitative approximation for the repertoire's stability, but does not take into account variation of call type usage (as defined by GMM clustering of the latent space). *Figure 4B*, shows the normalized usage of each cluster type over development for each family. Cluster usages during the period of 'full family, shared recording days' (postnatal days beneath the purple bars) are stable across days within families – as is apparent by the horizontal striations in the plot – although each family maintains this stability through using a unique set of call types. This is addressed empirically in *Figure 4C*, which shows clearly separable PCA projections of the cluster usages shown in *Figure 4B* (purple days, concatenated into a 45 day x 70 cluster matrix). Finally, we computed the pairwise Maximum Mean Discrepancy (MMD) between latent distributions of vocalizations from individual recording days for each of the families (*Figure 4D*). This shows that across-family repertoire differences are substantially larger than within-family differences. This is visualized in a multidimensional scaling projection of the MMD matrix in *Figure 4E*.

## Transition structure, but not emission structure, shows family specific differences

To assess whether temporal features also harbor family differences, we analyzed vocalization emission over a range of ethologically relevant timescales. First, we summed the total vocal emission for each hour of the day over the entire recording period, which revealed a diurnal activity pattern that was similar across the three families recorded (*Figure 5A*). We then analyzed a shorter time scale, the inter-vocalization-interval. The distribution of intervals between subsequent vocalizations is broad, with some vocalizations occurring rapidly after one another (e.g. within tens to hundreds of milliseconds) and others separated by many seconds. The majority of vocalizations occurred in bouts (58.5±0.9%), which we extracted using two criteria: (1) vocalizations within a bout display inter-vocalization-interval of <2 s, and (2) a bout contains at least 5 vocalizations (based on *Rose et al., 2021*). The distribution of bout durations, inter-vocalization-intervals, and vocalization durations for each family are highly overlapping and contain the same peaks (*Figure 5B–D*), suggesting that the temporal structure of vocal emission does not vary by family. Vocalization bouts show striking structure in vocal type sequencing (*Figure 5E, F*), therefore we next assessed whether vocal cluster sequencing varied by family. Vocal cluster transition matrices revealed a strong self-transition preference for all vocal clusters across families (*Figure 5G*); however, the proportion usage of different transitions (including self-transitions) drastically varied by family (*Figure 5H*).

To determine whether differences in 1 gram structure contribute to differences in the transition (2 gram) structure, we performed a number of controls. Although subtle, vertical streaks are clearly

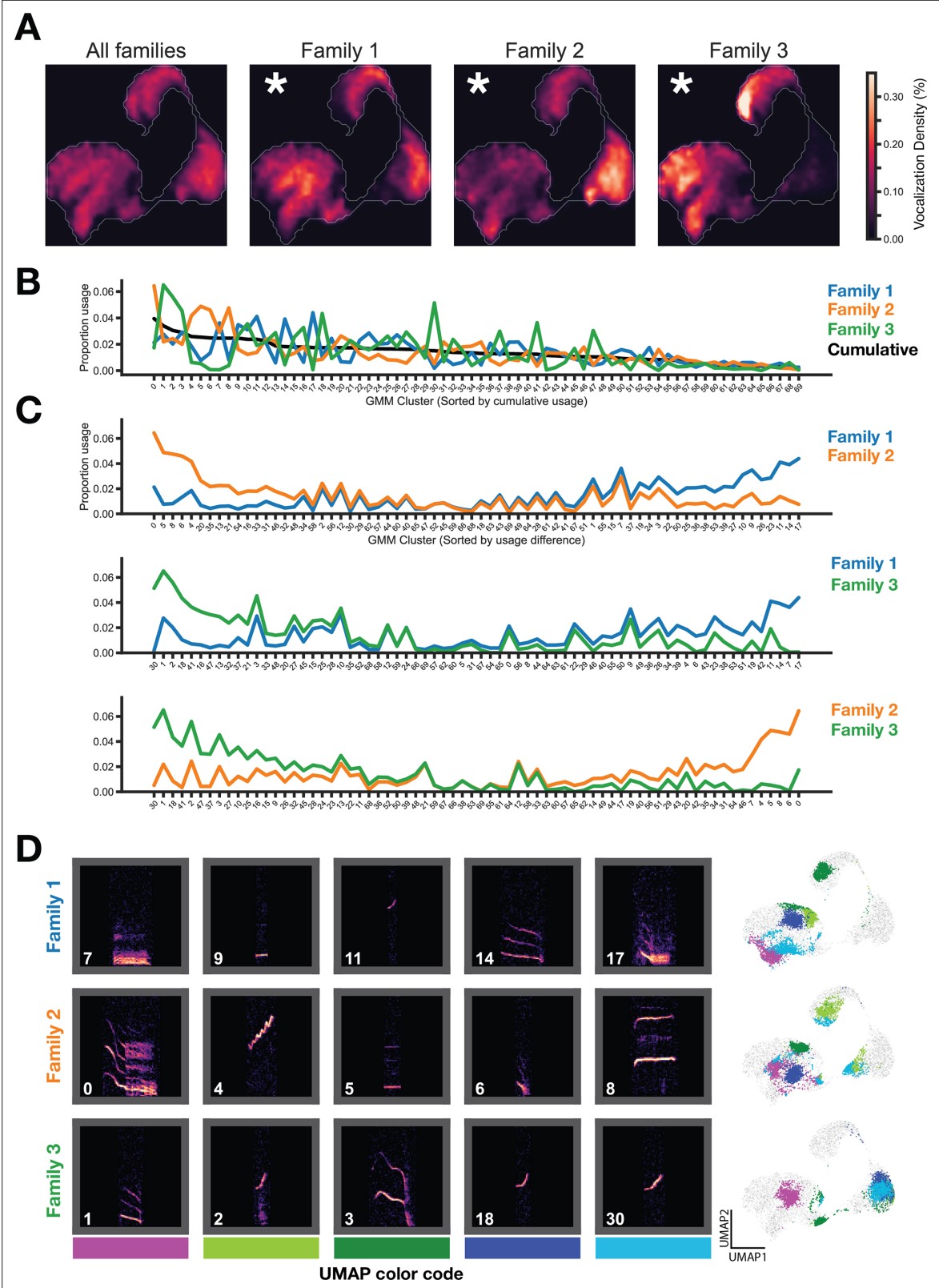

**Figure 3.** Family specific vocal usage. (**A**) UMAP probability density plots (axes same as *Figure 2B*) show significant differences between family repertoires (p<0.01, MMD permutation test on latent space; see Methods). (**B**) GMM vocal cluster usage by family. Clusters sorted by cumulative usage across all families. Families show distinct usage patterns of different vocal clusters. (**C**) Clusters are resorted by the usage difference between families. (**D**) Spectrogram examples from top differentially used clusters (left) and location of clusters in embedding space (right).

*Figure 3 continued on next page*

*Figure 3 continued*

The online version of this article includes the following figure supplement(s) for figure 3:

**Figure supplement 1.** Pup removal biases vocal repertoire usage.

**Figure supplement 2.** Acoustic features for GMM clusters.

present in shuffled transition matrices that correspond to 1 gram usages (*Figure 5—figure supplement 1A, B*). Given the shuffled data structure, we sought to determine whether the observed transition probabilities differed significantly from chance levels. We randomly shuffled label sequences 1000 times independently for each family to generate a null transition matrix distribution. Using these null distributions and the observed transition probabilities, we computed a p-value for each transition using a one-sample t-test and created a binary transition matrix indicating which transitions happen above chance levels (*Figure 5—figure supplement 1C*, black pixels, $P \leq 0.05$ after post hoc Benjamini-Hochberg multiple comparisons correction). As is made clear in *Figure 5—figure supplement 1C*,

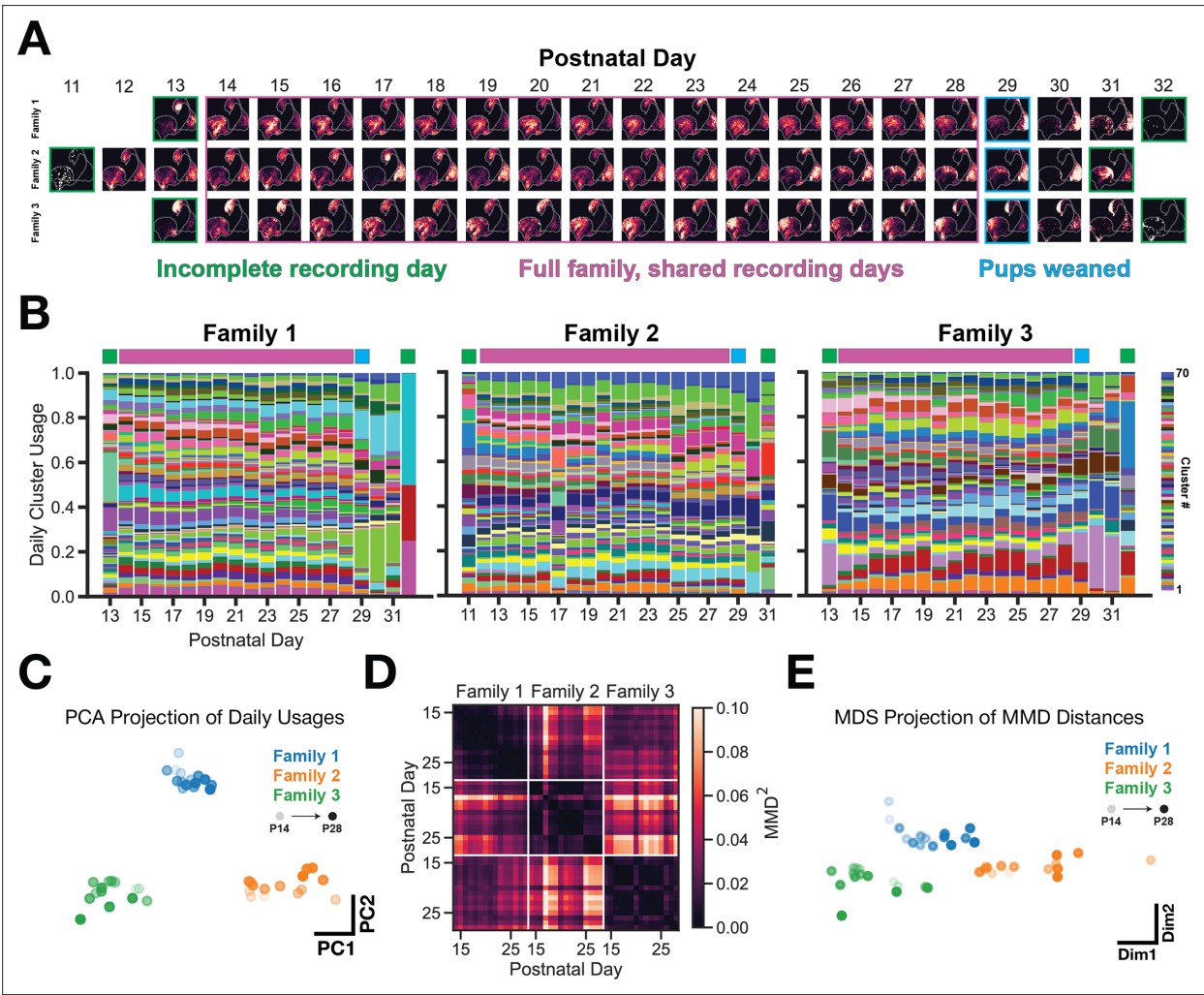

**Figure 4.** Vocal usage differences remain stable across days of development. (**A**) UMAP probability density plots for each day of the recording, across families. Purple box indicated recording days that are shared across families. These days are used for subsequent analyses in **C-E**. (**B**) GMM vocal cluster usage per day. Usages are normalized on a per-day basis. A unique color is used for each cluster type. (**C**) PCA projection of daily usages within the purple (shared recording days) period showing that families use a unique subset of clusters stably across days. (**D**) Maximum Mean Discrepancy (MMD) distance between VAE latent distributions of vocalizations between days and across families. (**E**) Multidimensional scaling projection of MMD matrix from (**D**). Family vocal repertoires are distinct and remain so across days.

The online version of this article includes the following figure supplement(s) for figure 4:

**Figure supplement 1.** Family specific cluster usages do not depend on GMM cluster size.

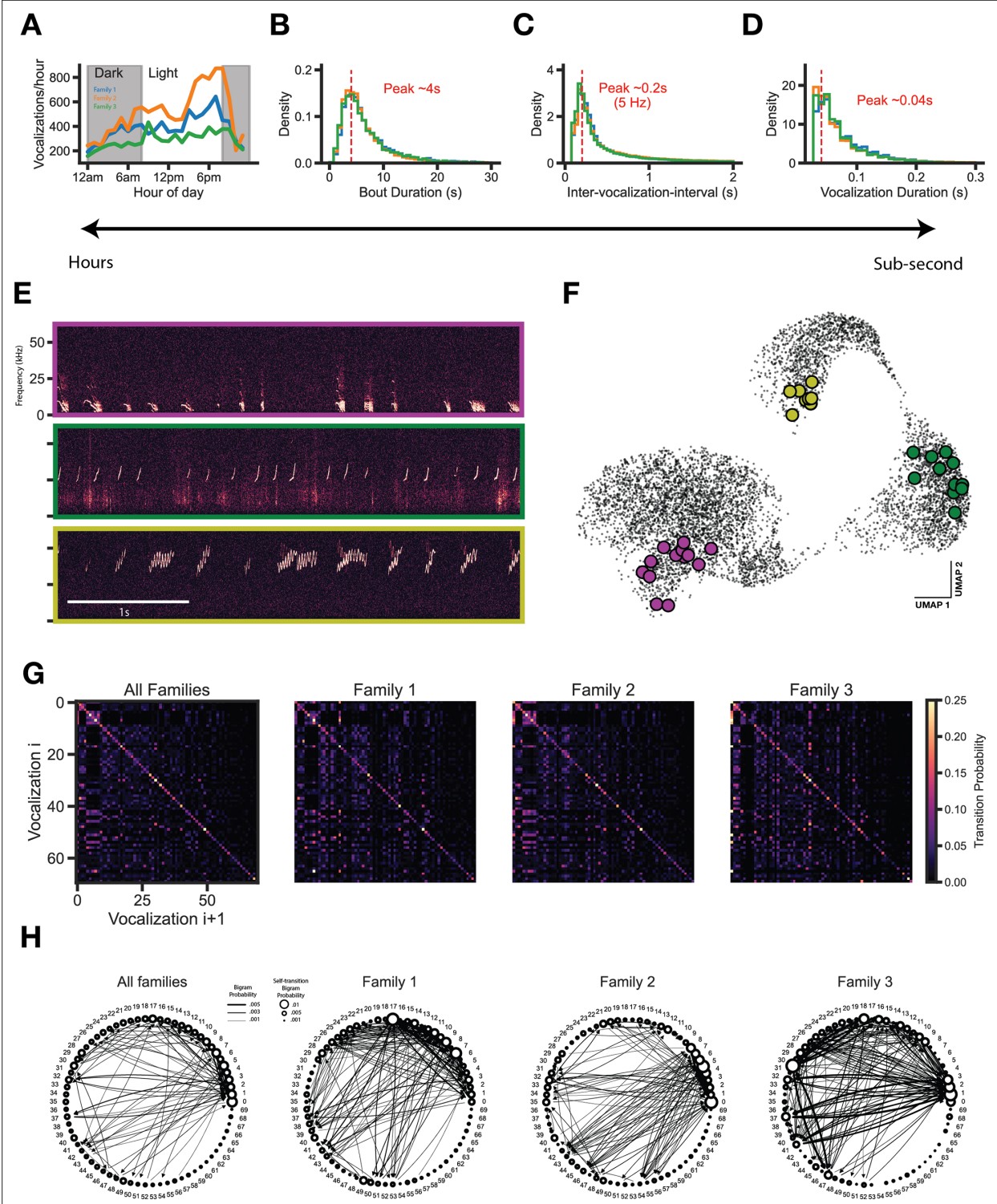

**Figure 5.** Transition structure, but not emission structure, shows family specific differences. (**A**) Vocalizations are emitted in a diurnal cycle. (**B**) Vocalizations consistently occur in seconds-long bouts across families. (**C**) Vocalization intervals (onset-to-onset) are consistent across families. (**D**) Vocalization durations are consistent across families. (**E**) Raw data examples of bouts. (**F**) Bouts typically occupy a similar area of vocal space. (**G**) Vocal cluster transition matrix. Vocalizations strongly favor self-transition. (**H**) Bigram probability graph. Self and other vocalization transition tendencies show family specific transitions (edges > 0.001 usage shown).

The online version of this article includes the following figure supplement(s) for figure 5:

**Figure supplement 1.** Vocalization transitions are non-random and family specific.

most transitions for each family occur significantly above chance levels, despite the inherent 1 gram structure. Moreover, by looking at transitions from a highly usage cluster type used roughly the same proportion across families (cluster 12), we show that families arrange the same sets of vocal clusters into unique sequences (*Figure 5—figure supplement 1D*). We believe that this provides compelling evidence that the 1 gram structure does not change the interpretation of the main claim that transition structure varies by family.

## Discussion

Understanding the neural mechanisms that support natural behaviors depends upon our ability to quantify specific actions over a range of ethologically relevant contexts and timescales (*Miller et al., 2022*). In principle, this requires continuous, undisturbed, and longitudinal recording that takes place in nature or a naturalistic context. This need has led to the emergence of powerful video tools for long-term monitoring and machine-learning based analyses (*Datta et al., 2019*; *Pereira et al., 2020*; *Shemesh and Chen, 2023*). In contrast, most studies of natural behavior have not acquired and analyzed acoustic information over prolonged periods, or from a socially intact cohort. Therefore, to characterize vocal communication in a canonical social group, we obtained continuous audio recordings from three separate Mongolian gerbil families over a 20-day period (*Figure 1*). By expanding the recording duration, and permitting animals undisturbed interaction with their family unit, we sought to capture a more diverse vocal repertoire, and to determine whether vocal attributes were associated with family identity.

Capitalizing on advances in computational bioacoustics, which aid in the characterization of complex and high-dimensional vocal behavior (*Sainburg et al., 2020*; *Sainburg and Gentner, 2021*; *Goffinet et al., 2021*), we extracted vocalization spectrograms and used a VAE to perform unsupervised analysis of a large number of familial gerbil vocalizations (n=583,237). At least one new vocal type and numerous multisyllabic vocal types were discovered using this approach (*Figure 2*). Also, we provide evidence that family structure is necessary to elicit the full vocal repertoire (*Figure 3—figure supplement 1*). These findings underscore the advantage of a longitudinal naturalistic approach, and suggest that further elaborations (e.g. providing a larger-scale naturalistic environment) could reveal new aural communication behaviors.

Social vocalizations can convey pivotal information about an animal's identity. For example, female macaques learn to recognize the vocalizations of their own offspring during the second postnatal week, and retain this ability for at least 6 months (*Jovanovic et al., 2000*; *Shizawa et al., 2005*). Similarly, kittens learn their mother's vocalizations, and Australian sea lions can recall their mother's voice up to 2 years after weaning (*Pitcher et al., 2010*; *Szenczi et al., 2016*). Furthermore, the meaning of vocal cues are often learned through long-term social experience. For example, when exposed to a chicken maternal call during development, socially reared mallard ducklings come to prefer it over their own species' call (*Gottlieb, 1993*). Similarly, wood ducklings must be exposed to sibling vocalizations in order to remain selectively responsive to its mother's assembly call (*Gottlieb, 1983*). Horseshoe bats, naked mole rats, and dolphins each model their calls based on early social experience (*Jones and Ransome, 1993*; *Fripp et al., 2005*; *Favaro et al., 2016*; *Barker et al., 2021*). Finally, rodent vocalizations can also harbor information about the individual identity and colony membership of the vocalizer (*Barker et al., 2021*). In fact, research on song learning in zebra finches shows that a reward learning mechanism may support the transmission of vocal repertoires: exposure to a live singing tutor, but not song playback, selectively activates dopamine neurons in the juvenile periaqueductal gray which is thought to mediate learning (*Tanaka et al., 2018*). Therefore, there is a compelling rationale for exploring the diversity of vocalizations within the gerbil family social group, and to pursue the underlying neural mechanisms in the future.

To address whether gerbils also exhibit family specific vocal features, we compared GMM-labeled vocal cluster usages across the three recorded families and showed differences in vocal type usage (*Figure 3*). Although we chose 70 clusters for our analyses, the general finding was robust across a wide range of GMM clusters (*Figure 4—figure supplement 1*). The differences in this study align with the definition of human vocal dialect, which is a regional or social variety of language that can differ in pronunciation, grammatical, semantic and/or language use differences (*Henry et al., 2015*). This definition of dialect is inclusive of both pronunciation differences (e.g. a Bostonian's characteristic pronunciation of 'car' as 'cah') and usage differences (e.g. a Bostonian's preferential usage of the

words 'Go Red Sox' vs. a New Yorker's preferential usage of the words 'Go Yankees'). In our case, vocal clusters can be rarely observed in some families yet highly overexpressed in others (e.g. analogous to language usage differences in humans), or highly expressed in both families, but contain subtle spectrotemporal variations (*Figure 3D*, Family 1 cluster 11 vs. Family 3 clusters 2, 18, 30; for example analogous to pronunciation differences in humans). Like another fossorial species, the naked mole-rat, it is possible that gerbils may also possess the ability to acquire family specific vocal behavior through experience (*Barker et al., 2021*). Unlike the naked mole-rat which showed the presence of a colony-specific vocal dialect in a single vocal type, the soft chirp, we show that fine spectrotemporal variations in multiple different gerbil vocal types could harbor dialects (*Figure 3D*).

The described family differences collapse data from multiple days into a single comparison; however, it is possible that factors such as vocal development and/or high usage of particular vocal types during specific periods of the recording could explain family differences. Therefore, we took advantage of the longitudinal nature of our dataset to assess whether repertoire differences remain stable across time. First, we visualized vocal repertoire usage across days as either UMAP probability density maps (*Figure 4A*) or daily GMM cluster usages (*Figure 4B*). Though qualitative, one can appreciate that family repertoire usage remains stable across days and appears to differ on a consistent daily basis across families. To formally quantify this, we first projected GMM cluster usages from *Figure 4B* into PC space and show that family GMM cluster usage patterns are highly separable, regardless of postnatal day (*Figure 4C*). If families had used a more overlapping set of call types, then the projections would have appeared intermixed. Next, we performed a cluster-free analysis by computing the pairwise MMD distance between VAE latent distributions of vocalizations from each family and day (*Figure 4D*). This analysis shows very low MMD values across days within a family (i.e. the repertoire is highly consistent with itself), and high MMD values across families/days (greater than would be expected by chance; see shuffle control in *Figure 2—figure supplement 1D*). The relative differences in this matrix are made clear in *Figure 4E*, which provides additional evidence that family vocal repertoires remain stable across days and are consistently different from other families. Taken together, we believe that this is compelling evidence that differences in vocal repertoires between families are not driven by dominating call types during specific phases in the recording period; rather, families consistently emit characteristic sets of call types across days. This opens up the possibility to assess repertoire differences over much shorter time periods (e.g. 24 hr) in future studies.

Vocalization emission statistics and behavioral syllable transition patterns can signify differences between groups of animals (*Castellucci et al., 2018*; *Wiltschko et al., 2015*; *Markowitz et al., 2018*). Therefore, it is possible that vocal emission patterns or vocal cluster transition patterns may be family specific. To address this, we first compared vocalization emission rates over multiple ethologically relevant timescales, which revealed highly consistent emission patterns across families (*Figure 5A–D*). First, we observed that vocal emission follows a diurnal pattern, with peaks of activity in the morning and afternoon. This result complements prior work in gerbils showing diurnal activity patterns in gerbil groups for non-vocal behaviors (*Pietrewicz et al., 1982*), but extends our understanding to vocal behavior. Vocalizations are rarely emitted in isolation. Rather, they are emitted in sequences ('bouts') with a modal duration of 4 s and a duration distribution that does not vary between families. These emission statistics are somewhat consistent with the common phoneme rate in humans (*Edwards and Chang, 2013*; *Ding et al., 2017*). Also, the distributions of inter-vocalization interval and vocalization duration did not differ between families. Taken together, the temporal emission structure is highly consistent across families and suggests that these features are likely not exploited for kinship identification. However, this does not rule out the possibility that the sequential organization of vocalizations could vary. Vocalization bouts (*Figure 1C*, *Figure 5E–F*) show that temporal sequencing of vocalization clusters is non-random and has a compelling transition structure with potential to vary across families. To formally quantify this, we calculated vocalization transition matrices for each family, which revealed that all families strongly favor vocalization self-transitions (*Figure 5G*), although hinted that non self-transitions (off-diagonal) vary by family. To visualize this, we generated bigram transition graphs of highly expressed vocalization transitions, which provides evidence that vocalization transition structure varies by family (*Figure 5H*, *Figure 4—figure supplement 1*). Importantly, families arrange the same sets of vocal clusters into unique sequences (*Figure 4—figure supplement 1D*). There are limitations to this study that deserve consideration. First, a fully realized assessment of vocalization usage should be integrated with continuous sub-second synchronized videographic data from which

one can extract animal pose estimation and behavioral categorization. For example, such data could allow us to control for the total number and type of social interactions, which may explain differences in the amount or usage of specific syllables. Second, although we used four microphones, it was not possible to localize the majority of vocalizations with sufficient spatial resolution (using Mouse Ultrasonic Source Estimation software; *Neunuebel et al., 2015*). To properly address whether individual animals emit a unique vocalization repertoire, we will require significant advances in the field of computational bioacoustics. In anticipation of future research in this area, we have computed acoustic features of vocalizations in each of the GMM clusters as a reference (*Figure 3—figure supplement 2*).

Although we were not able to attribute vocalizations to individual family members, we did seek to determine the importance of family structure by comparing audio recordings before and after removal of the pups at P30. The results show a clear effect of family integrity, and the sudden reduction of sonic calls following pup removal (*Figure 3—figure supplement 1*) could suggest that these vocalizations are produced selectively by pups. However, there is ample evidence that adult gerbils also produce sonic vocalizations. For example, a number of low-frequency call types are used by adults during a range of social interactions (*Rübsamen et al., 2012*; *Furuyama et al., 2022*), some of which are similar to a low-frequency call type used by pups (*Silberstein et al., 2023*). Vocalization patterns of developing gerbils depend on isolation or staged interactions. Thus, when gerbil pups are recorded during isolation, ultrasonic vocalization rate declines and sonic vocalizations increase for animals that are in a high arousal state (*De Ghett, 1974*; *Silberstein et al., 2023*). As gerbils progress from juvenile to adolescent development (P17-55) a significant increase in ultrasonic vocalization rate is observed during dyadic social encounters, with a distinct change in usage pattern that depends upon the sex of each animal (*Holman and Seale, 1991*; *Holman et al., 1995*). The development of vocalization types has been assessed in another member of the Gerbillinae subfamily, called fat-tailed gerbils (Pachyuromys duprasi), during isolation and handling. Here, the number of ultrasonic vocalization syllable types increase from neonatal to adult animals (*Zaytseva et al., 2019*), while some very low frequency sonic call types were rarely observed after P20 (*Zaytseva et al., 2020*). By comparison, mouse syllable usage changes during development, but pups produced 10 of the 11 syllable types produced by adults (*Grimsley et al., 2011*). In summary, our understanding of the maturation of vocalization usage remains limitted by our inability to obtain longitudinal data from individual animals within their natural social setting. For example, when recorded in their natural environment, chimpanzees display a prolonged maturation of vocalization complexity, such as the probability of a unique utterance in a sequence, with the greatest changes occuring when animals begin to experience non-kin social interactions (*Bortolato et al., 2023*).

These results reveal that Mongolian gerbil families possess a rich repertoire of vocalizations used during day-to-day communication. Our findings indicate that long-term behavioral monitoring of a core social unit (i.e. the family) reveals richer vocal behavior than has previously been reported in the species. Leveraging unsupervised machine learning to quantify vocalizations, we reveal family-specific vocalization usage and transition structure. Taken together, these findings establish the Mongolian gerbil as a useful model organism for studying the neurobiology of vocal interactions in complex social groups.

## Methods

### Experimental animals

Three gerbil families (*Meriones unguiculatus*, n=6 per family: 2 adults, 4 pups) were used in this study (Charles River). All procedures related to the maintenance and use of animals were approved by the University Animal Welfare Committee at New York University (protocol 2020–1112), and all experiments were performed in accordance with the relevant guidelines and regulations.

### Audio recording

Four ultrasonic microphones (Avisoft CM16/CMPA48AAF-5V) were synchronously recorded using a National Instruments multifunction data acquisition device (PCI-6143) via BNC connection with a National Instruments terminal block (BNC-2110). The recording was controlled with custom python scripts using the NI-DAQmx-python library version 0.5.7; (https://github.com/ni/nidaqmx-python; *National Instruments, 2017*) which wrote samples to disk at a 125 kHz sampling rate. In total, 13.084

TB of raw audio data were acquired across the three families. For further analyses, the four-channel microphone signals were averaged to create a single-channel high-fidelity audio signal.

## Audio segmentation

Audio was segmented by amplitude thresholding using the Autoencoded Vocal Analysis (AVA) python package (see *Goffinet et al., 2021*). First, sound amplitude traces are calculated by computing spectrograms from raw audio, then summing each column of the spectrogram. The 'get_onset_offsets' function, which performs the segmenting, requires the selection of a number of parameters which affect segmenting performance. The following values were tuned via an interactive procedure which validated that the segmenting could detect low amplitude vocalizations and capture individual vocal units apparent by eye:

```
seg_params = {
'min_freq': 500 # minimum frequency
'max_freq': 62500, # maximum frequency
'nperseg': 512, # FFT'noverlap': 256, # FFT
'spec_min_val': -8, # minimum STFT log-modulus
'spec_max_val': -7.25, # maximum STFT log-modulus
'fs': 125000, # audio sample rate
'th_1': 2, # segmenting threshold 1
'th_2': 5, # segmenting threshold 2
'th_3': 2, # segmenting threshold 3
'min_dur':0.03, # minimum syllable duration (s)
'max_dur':0.3, # maximum syllable duration (s)
'smoothing_timescale': 0.007, # amplitude
'softmax': False, # apply softmax to the frequency bins to calculate
amplitude
'temperature':0.5, # softmax temperature parameter
'algorithm': get_onsets_offsets
}
```

Sound onsets are detected when the amplitude exceeds 'th_3' (black dashed line, *Figure 1C*), and sound offset occurs when there is a subsequent local minimum for example amplitude less than 'th_2' (gray dashed line, *Figure 1C*), or 'th_1' (black dashed line, *Figure 1C*), whichever comes first. In this specific use case, th_2 (5) will always come before th_1 (2), therefore the gray dashed line will always be the offset. A subsequent onset will be marked if the sound amplitude crosses th_2 or th_3, whichever comes first. For example, the first sound event detected in *Figure 1C* shows the sound amplitude rising above the black dashed line (th_3) and marks an onset. Subsequently, the amplitude trace falls below the gray dashed line (th_2) and an offset is marked. Finally, the amplitude rises above th_2 without dipping below th_3 and an onset for a new sound event is marked. Had the amplitude dipped below th_3, a new sound event onset would be marked when the amplitude trace subsequently exceeded th_3 (e.g. between sound event 2 and 3, *Figure 1C*). The maximum and minimum syllable durations were selected based on published duration ranges of gerbil vocalizations (*Rübsamen et al., 2012*; *Kobayasi and Riquimaroux, 2012*).

## Vocalization extraction

We computed the spectral flatness of each detected sound event using the python package librosa (https://github.com/librosa; *McFee et al., 2024*). Consistent with prior literature (*Castellucci et al., 2016*), we used a threshold on spectral flatness to separate putative vocal and non-vocal sounds. This threshold value was determined empirically, by calculating the false positive vocalization rate (*Figure 1—figure supplement 1*) of groups of randomly sampled vocalizations. For each spectral flatness value in *Figure 1—figure supplement 1B*, 100 randomly sampled vocalization spectrograms less than the working threshold value were assembled into 10x10 grids and visually inspected for false positives (e.g. non-vocal sounds; *Figure 1—figure supplement 1C*). This procedure was repeated 10 times for spectral flatness thresholds of 0.1, 0.15, 0.2, 0.25, 0.3, 0.35, and 0.4. We quantified the false

positive vocalization rate for each threshold value and selected 0.3, which had a 5.5+/-1.96% false positive rate.

## Variational autoencoder training

Extracted vocalizations were converted to 128x128 pixel spectrograms using the 'process_sylls' function from AVA with the following preprocessing parameters:

```
preprocess_params = {
'get_spec': get_spec, # spectrogram maker
'max_dur': 0.3, # maximum syllable duration'min_freq': 500, # minimum
frequency
'max_freq': 62500, # maximum frequency
'nperseg': 512, # FFT
'noverlap': 256, # FFT
'spec_min_val': -8, # minimum log-spectrogram value
'spec_max_val': -5, # maximum log-spectrogram value
'fs': 125000, # audio sample rate
'mel': False, # frequency spacing, mel or linear
'time_stretch': True, # stretch short syllables?
'within_syll_normalize': False, # normalize spectrogram values on a
                        # spectrogram-by-spectrogram basis
'max_num_syllables': None, # maximum number of syllables per directory
'sylls_per_file': 100, # syllable per file
'real_preprocess_params': ('min_freq', 'max_freq',
    'spec_min_val','spec_max_val', 'max_dur'), # tunable parameters
'int_preprocess_params': ('nperseg','noverlap'), # tunable parameters
'binary_preprocess_params': ('time_stretch', 'mel',
    'within_syll_normalize'), # tunable parameters
}
```

A VAE was trained for 50 epochs using a model precision of 40. We removed additional false positive vocalizations by inspecting a 2D UMAP embedding of the VAE latent space and removing UMAP clusters containing non-vocal sounds from further analysis.

## Gaussian mixture model

GMMs were fit to cluster VAE latent feature vectors. To reduce computation time, we fit the model on 7 of 32 VAE latents (*Figure 2—figure supplement 1E*), as these explained 99.5% of the variance in the original feature space. The model was implemented in Stan (https://mc-stan.org/cmdstanpy), however similar clustering results were achieved using the scikit-learn Gaussian Mixture model class with a diagonal covariance matrix (https://scikit-learn.org/stable/modules/generated/sklearn.mixture.GaussianMixture.html). We fit the model using stochastic variational inference, an approximate Bayesian inference technique that recasts the task of learning a posterior distribution as an optimization problem and enables vast speedups (*Hoffman et al., 2013*). GMMs typically assume that the whole population selects clusters with the same probabilities, however we modified this assumption to allow, although not enforce, the model to learn different cluster usage patterns for each family. Specifically, we used the following model:

Let $D$ be the dimensionality of the VAE latents used (in our case, $D$=7) and $K$ be the number of clusters. Denote our parameters by:

Mixture means ($\beta$) for cluster $j$: $\beta_j \in \mathbb{R}^D$
Mixture covariance matrix ($\Sigma$) for cluster $j$: $\Sigma_j = [diag(\sigma_j)]^2$, for $\sigma_j \in \mathbb{R}^D$
Cluster usage probabilities for cohort $i$: $\theta_i \in \mathbb{R}^K$, with $\sum_{j=1}^{K} \theta_{i,j} = 1$
Cluster assignment for vocalization $k$ of cohort $i$: $z_{ik} \in \{1, ..., K\}$

We selected our hyperparameters according to Stan's guidelines for weakly informative priors, yielding the model:

Mixture means for cluster j: $\beta_j \sim Normal_D \left(0, 5\right)$
Mixture standard deviations ($\sigma$) for cluster j: $\sigma_j \sim HalfNormal_D \left(3\right)$
Cluster usage probabilities for cohort $i$: $\theta_i \sim Dirichlet \left(1, ..., 1\right)$
Cluster assignment for vocalization k of cohort i: $z_{ik} \, Categorical \left(\theta_i\right)$

VAE feature embedding for vocalization k of cohort i: $x_{ik} \, Normal \left(\beta_{(z_{ik})}, \Sigma_{(z_{ik})}\right)$

To select the number of clusters, *K*, we held out 25% of our data, trained models with varying values for *K*, and calculated the log probability of seeing the held-out data under each model (*Figure 2—figure supplement 1C*). Using the elbow method, we determined that ~70 clusters was a reasonable selection for K. Previous work documenting the Mongolian gerbil repertoire (*Rübsamen et al., 2012*; *Kobayasi and Riquimaroux, 2012*) has revealed ~12 vocalization types that vary with social context. It is likely that we are capturing these ~12 (plus a few more, as illustrated in *Figure 2C*) as well as individual or family-specific variations of some call types. Although the number of discrete call types is likely less than 70, it is plausible that variation due to vocalizer identity pushes some calls into unique clusters. This idea is supported by the fact that both naked mole rats and Mongolian gerbils have been shown to exhibit individual-specific variation in vocalizations, though only in single call types (Figure 1 from *Barker et al., 2021*; Table I from *Nishiyama et al., 2011*). Importantly, the core result is not affected by cluster size (*Figure 5—figure supplement 1*).

## Maximum mean discrepancy permutation test

Clustering analyses are notoriously challenging (*Kleinberg, 2002*). Thus, we performed a complementary analysis to investigate whether different gerbil families utilize different vocal repertoires. In particular, we pursued an approach that makes no assumptions about the number, character, or even existence of vocalization clusters.

Specifically, we used Maximum Mean Discrepancy (MMD) to quantify the difference between two latent distributions of vocalizations. This test considers two sets of observed data points (e.g. N vocalizations from Family 1 and N vocalizations from Family 2), which are assumed to be independent and identically distributed random variables from underlying probability distributions, and returns a distance metric corresponding to the equality of the two distributions (*Gretton et al., 2012*). Lower values suggest distributions are more similar and higher values suggest distributions are more dissimilar. We investigated the null hypothesis that the gerbil families used the same vocal repertoire—that is the probability distribution over VAE latent space for each family was identical, corresponding to a $MMD^2$ distance of zero. To test this null hypothesis, we computed the $MMD^2$ distance between the empirical distributions of family pairs in batches of 1000 randomly sub-sampled vocalizations. This yielded a histogram of empirically observed $MMD^2$ distance values for each family pair, which we compared a null distribution generated by randomly permuting the family label attached to each vocalization. The empirically observed $MMD^2$ distances were much higher than the shuffle control, favoring the alternative hypothesis that gerbil families utilize distinct syllable usage statistics (*Figure 2—figure supplement 1*).

## Transition analysis

Vocalization transition sequences were generated by concatenating vocal cluster labels chronologically for each family and calculating the number of transitions for all possible transition types. The resulting transition matrix was normalized such that each row sums to 1, thus reflecting the probability that vocalization $i$ transitions to vocalization $i + 1$, that is $p_i \left(j\right)$ (*Figure 5G*). The transition matrix used to generate the bigram probability graph in *Figure 5H* was normalized such that edge and node widths correspond to the probability of each vocalization pair, that is $p \left(i, j\right)$ (*Shannon, 1948*).

## Acoustic feature calculations

First, raw audio from the most probable vocalization samples (n=100) from each vocal cluster were extracted. Next, using the VocalPy (*Nicholson, 2023*) 'similarity_features' function (a python implementation of the Sound Analysis Pro Sound Analysis Tools library: http://soundanalysispro.com/matlab-sat), the following acoustic features were calculated: fundamental frequency (pitch), amplitude, entropy, frequency modulation, goodness of pitch. In addition to these features, spectral flatness was computed using librosa (https://librosa.org/doc/latest/generated/librosa.feature.spectral_flatness.html), and duration was computed from the raw audio itself. Finally, start and stop frequencies

were computed by taking the median fundamental frequency within the first third and last third (time) of each vocalization, respectively.

The following spectrogram parameters were used: nfft = 512, noverlap (a.k.a hop_length)=256, Fs = 125000, min_freq = 65, max_freq = 62,500. Features are computed on a spectrogram frame-by-frame basis. Single values for each vocalization were extracted by taking the median acoustic feature value across all spectrogram frames. The single exception to this was spectral flatness (to remain consistent with the spectral flatness calculation used for amplitude thresholding), which took the mean across all spectrogram frames and used the following spectrogram parameters: n_fft = 256, hop_length = 128, win_length = 256, center = False, power = 2.0.

Detailed description of the units associated with each feature are located here: http://soundanaly-sispro.com/manual/chapter-4-the-song-features-of-sap2. Code to compute acoustic features is available on GitHub (https://github.com/ralphpeterson/gerbil-vocal-dialects, copy archived at *Peterson, 2024*).

## Acknowledgements

We thank Michael Long, Christine Constantinople, and members of the Sanes, Schneider, and Williams laboratories for helpful discussions and feedback on the study. We thank Nicholas Jourjine and David Nicholson for advice on acoustic feature calculations. This work was supported by National Institutes of Health Training Program in Computational Neuroscience T90DA059110 (REP), National Institutes of Health Training Program in Neuroscience 5T32MH096331 (REP), National Institute on Deafness and Other Communication Disorders at the National Institute of Health (R01DC020279 to DHS, R01DC018802 to DMS with supplement to REP, and R34DA059513 to AHW, DMS, and DHS), a Career Award at the Scientific Interface from the Burroughs Wellcome Fund (DMS), and fellowships from the Searle Scholars Program, the Alfred P Sloan Foundation, and the McKnight Foundation (DMS). DMS is a New York Stem Cell Foundation - Robertson Neuroscience Investigator.

## Additional information

### Funding

| Funder | Grant reference number | Author |
|---|---|---|
| National Institutes of Health | T90DA059110 | Ralph E Peterson |
| National Institutes of Health | 5T32MH096331-10 | Ralph E Peterson |
| National Institute on Deafness and Other Communication Disorders | R01DC020279 | Dan H Sanes |
| National Institute on Deafness and Other Communication Disorders | R01DC018802 | David M Schneider |
| National Institute on Deafness and Other Communication Disorders | R34DA059513 | Alex H Williams David M Schneider Dan H Sanes |
| Burroughs Wellcome Fund | Career Award at the Scientific Interface | David M Schneider |
| Searle Scholars Program | | David M Schneider |
| Alfred P. Sloan Foundation | | David M Schneider |
| McKnight Foundation | | David M Schneider |
| New York Stem Cell Foundation | Robertson Neuroscience Investigator | David M Schneider |

| Funder | Grant reference number | Author |
|--------|------------------------|--------|

The funders had no role in study design, data collection and interpretation, or the decision to submit the work for publication.

## Author contributions

Ralph E Peterson, Conceptualization, Resources, Data curation, Software, Formal analysis, Supervision, Validation, Investigation, Visualization, Methodology, Writing – original draft, Project administration, Writing – review and editing; Aman Choudhri, Formal analysis, Visualization, Writing – review and editing; Catalin Mitelut, Conceptualization, Software, Methodology; Aramis Tanelus, Resources, Software, Methodology; Athena Capo-Battaglia, Data curation; Alex H Williams, Resources, Supervision, Writing – review and editing; David M Schneider, Dan H Sanes, Conceptualization, Resources, Supervision, Funding acquisition, Investigation, Writing – review and editing

## Author ORCIDs

Ralph E Peterson ⓘ https://orcid.org/0000-0002-2692-5955
Catalin Mitelut ⓘ https://orcid.org/0000-0003-0471-9816
Dan H Sanes ⓘ https://orcid.org/0000-0002-3783-6165

## Ethics

Three gerbil families (Meriones unguiculatus, n=6 per family: 2 adults, 4 pups) were used in this study (Charles River). All procedures related to the maintenance and use of animals were approved by the University Animal Welfare Committee at New York University, and all experiments were performed in accordance with the relevant guidelines and regulations. (protocol 2020-1112).

Reviewer #1 (Public review): https://doi.org/10.7554/eLife.89892.3.sa1
Reviewer #2 (Public review): https://doi.org/10.7554/eLife.89892.3.sa2
Reviewer #3 (Public review): https://doi.org/10.7554/eLife.89892.3.sa3
Author response https://doi.org/10.7554/eLife.89892.3.sa4

# Additional files

## Supplementary files

MDAR checklist

## Data availability

Data analyzed in this study are freely available for download on Dryad (https://doi.org/10.5061/dryad.m905qfv68). Code is available on GitHub (https://github.com/ralphpeterson/gerbil-vocal-dialects, copy archived at *Peterson, 2024*).

The following dataset was generated:

| Author(s) | Year | Dataset title | Dataset URL | Database and Identifier |
|-----------|------|---------------|-------------|-------------------------|
| Peterson RE, Choudhri A, Mitelut C, Tanelus A, Capo-Battaglia A, Wililams A, Schneider A, Sanes D | 2024 | Data for: Unsupervised discovery of family specific vocal usage in the Mongolian gerbil | https://datadryad.org/stash/dataset/doi:10.5061/dryad.m905qfv68 | Dryad Digital Repository, 10.5061/dryad.m905qfv68 |

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
