## [Editor Report · eLife Assessment]

This **valuable** study provides an experimental paradigm and state-of-the-art analysis method for studying the existence of call types and transition differences among Mongolian gerbil families in a naturalistic environment. The analyses are **convincing**, with a thorough treatment of the acoustic data and a demonstration of the robustness of the observed effect across days. The work will likely be of interest to the auditory neuroscience and neuroethology communities.

---

## [Referee Report · Reviewer #1 (Public review)]

Summary:

This research offers an in-depth exploration and quantification of social vocalization within three families of Mongolian gerbils. In an enlarged, semi-natural environment, the study continuously monitored two parent gerbils and their four pups from P14 to P34. Through dimensionality reduction and clustering, a diverse range of gerbil call types was identified. Interestingly, distinct sets of vocalizations were used by different families in their daily interactions, with unique transition structures exhibited across these families. The primary results of this study are compelling, although some elements could benefit from clarification

Strengths:

Three elements of this study warrant emphasis. Firstly, it bridges the gap between laboratory and natural environments. This approach offers the opportunity to examine natural social behavior within a controlled setting (such as specified family composition, diet, and life stages), maintaining the social relevance of the behavior. Secondly, it seeks to understand short-timescale behaviors, like vocalizations, within the broader context of daily and life-stage timescales. Lastly, the use of unsupervised learning precludes the injection of human bias, such as pre-defined call categories, allowing the discovery of the diversity of vocal outputs.

Comments on the revised version:

(1) The authors have clarified the possible types of differences in the vocalizations of different families and discussed the potential contribution of the adult-pup difference.

(2) The authors have added the analysis in Figure 4 about the developmental changes in call types.

(3) The authors have analyzed the additional information in the 2-gram structure of the calls as evidence to apply the transition matrices to compare the families.

---

## [Referee Report · Reviewer #2 (Public review)]

Peterson et al., perform a series of behavioral experiments to study the repertoire and variance of Mongolian gerbil vocalizations across social groups (families). A key strength of the study is the use of a behavioral paradigm which allows for long term audio recordings under naturalistic conditions. This new experimental set-up results in the identification of additional vocalization types, not previously described the literature. In combination with state-of-the-art methods for vocalization analysis, the authors demonstrate that the distribution of sound types and the transitions between these sound types across three gerbil families is different. This is a highly compelling finding which suggests that individual families may develop distinct vocal repertories. One potential limitation of the study lies in the cluster analysis used for identifying distinct vocalization types. The authors use a Gaussian Mixed Model (GMM) trained on variational auto Encoder derived latent representation of vocalizations to classify recorded sounds into clusters. Through the analysis the authors identify 70 distinct clusters and demonstrate a differential usage of these sound clusters across families. While the authors acknowledge the inherent challenges in cluster analysis and provide additional analyses (i.e. maximum mean discrepancy, MMD), additional analysis would increase the strength of the conclusions. In particular, analysis with different cluster sizes would be valuable. An additional limitation of the study is that due to the methodology that is used, the authors can not provide any information about the bioacoustic features that contribute to differences in sound types across families which limits interpretations about how the animals may perceive and react to these sounds in an ethologically relevant manner.

The conclusions of this paper are well supported by data.

• Can the authors comment on the potential biological significance of the 70 sound clusters? Does each cluster represent a single sound type? How many vocal clusters can be attributed to a single individual? Similarly, can the authors comment on the intra-individual and inter-individual variability of the sound types within and across families?

• As a main conclusion of the paper rests on the different distribution of sound clusters across families, it is important to validate the robustness of these differences across different cluster parameters. Specifically, the authors state that "we selected 70 clusters as the most parsimonious fit". Could the authors provide more details about how this was fit? Specifically, could the authors expand upon what is meant by "prior domain knowledge about the number of vocal types...". If the authors chose a range of cluster values (i.e. 10, 30, 50, 90) does the significance of the results still hold?

• While VAEs are powerful tools for analyzing complex datasets in this case they are restricted to analysis of spectrogram images. Have the authors identified any acoustic differences (i.e. in pitch, frequency, other sound components) across families?

Following a revision of the manuscript the authors have taken many of these points under consideration and as a result have significantly improved the manuscript. Critically, they have now provided additional quantification that differences across family repertories are robust against cluster selection size.

---

## [Referee Report · Reviewer #3 (Public review)]

Summary:

In this study, Peterson et al. longitudinally record and document the vocal repertoires of three Mongolian gerbil families. Using unsupervised learning techniques, they map the variability across these groups, finding that while overall statistics of, e.g., vocal emission rates and bout lengths are similar, families differed markedly in their distributions of syllable types and the transitions between these types within bouts. In addition, the large and rich data are likely to be valuable to others in the field.

Strengths:

- Extensive data collection across multiple days in multiple family groups.

- Thoughtful application of modern analysis techniques for analyzing vocal repertoires.

- Careful examination of the statistical structure of vocal behavior, with indications that these gerbils, like naked mole rats, may differ in repertoire across families.

- Estimation of the stability of the effects across days.

Weaknesses:

- The work is largely descriptive, documenting behavior rather than testing a specific hypothesis.

- The number of families (N=3) is somewhat limited, though the authors have taken some care to examine the robustness of the findings.

---

## [Author Response]

The following is the authors’ response to the original reviews.

**Public Reviews:**

**Reviewer #1 (Public Review):**
Summary:This research offers an in-depth exploration and quantification of social vocalization within three families of Mongolian gerbils. In an enlarged, semi-natural environment, the study continuously monitored two parent gerbils and their four pups from P14 to P34. Through dimensionality reduction and clustering, a diverse range of gerbil call types was identified. Interestingly, distinct sets of vocalizations were used by different families in their daily interactions, with unique transition structures exhibited across these families. The primary results of this study are compelling, although some elements could benefit from clarificationStrengths:Three elements of this study warrant emphasis. Firstly, it bridges the gap between laboratory and natural environments. This approach offers the opportunity to examine natural social behavior within a controlled setting (such as specified family composition, diet, and life stages), maintaining the social relevance of the behavior. Secondly, it seeks to understand short-timescale behaviors, like vocalizations, within the broader context of daily and life-stage timescales. Lastly, the use of unsupervised learning precludes the injection of human bias, such as pre-defined call categories, allowing the discovery of the diversity of vocal outputs.Weaknesses:(1) While the notable differences in vocal clusters across families are convincing, the drivers of these differences remain unclear. Are they attributable to "dialect," call usage, or specific vocalizing individuals (e.g., adults vs. pups)? Further investigation, via a literature review or additional observation, into acoustic differences between adult and pup calls is recommended. Moreover, a consistent post-weaning decrease in the bottom-left cluster (Fig. S3) invites interpretation: could this reflect drops in pup vocalization?

Thank you for bringing up this point of clarification. Without knowledge of individual vocalizers, we are unable to rigorously assess pronunciation differences between individuals, however we can get a clear proxy for dialect through observing usage differences between families. We’ve added the following text (blue) in the Discussion to help clarify:

“To address whether gerbils also exhibit family specific vocal features, we compared GMM-labeled vocal cluster usages across the three recorded families and showed differences in vocal type usage (Figure 3). The differences in this study align with the definition of human vocal dialect, which is a regional or social variety of language that can differ in pronunciation, grammatical, semantic and/or language use differences (Henry et al., 2015). This definition of dialect is inclusive of both pronunciation differences (e.g. a Bostonian’s characteristic pronunciation of “car” as “cah”) and usage differences (e.g. a Bostonian’s preferential usage of the words “Go Red Sox” vs. a New Yorker’s preferential usage of the words “Go Yankees”). In our case, vocal clusters can be rarely observed in some families yet highly over-expressed in others (e.g. analogous to language usage differences in humans), or highly expressed in both families, but contain subtle spectrotemporal variations (Figure 3D, Family 1 cluster 11 vs. Family 3 clusters 2, 18, 30; e.g. analogous to pronunciation differences in humans).”

Indeed, our recordings obtained after pup removal could suggest that adults may use fewer low frequency calls (bottom left cluster in UMAP). However, this dataset does not permit a proper assessment of post-weaning pup calls. In fact, our results and the literature shows that adults are likely to use low frequency calls, but only during social interactions with pups or other adults. For example, Furuyama et al. 2022 describe a number of low frequency call types used by adults in agonistic social interactions, which look similar to a low frequency call type used by pups described in Silberstein et al. 2023. Similarly, Ter-Mikaelian et al. 2012 (their Figure 6) recorded several types of sonic vocalizations during adult social interaction. To our knowledge, it has not been shown whether gerbil pups and adults produce distinct call types. It is a challenging problem to solve, as animals placed in isolation (i.e. an experimental condition for which the identity of the vocalizer is known) vocalize infrequently and of the limited number they might emit, they do not use the full range of vocalizations described in the literature (RP personal observations). To properly address this question, one would need to elicit full use of the vocal repertoire through free social interaction, then attribute calls to individual vocalizers via sound source localization and/or head-mounted microphones — we are currently pursuing both of these technical challenges, but this is outside the scope of this manuscript.

Although the literature reflects the limitations discussed above, we have added a brief paragraph to the Discussion (limitations section) that addresses the reviewer’s question about the development of vocalizations:

“Although we were not able to attribute vocalizations to individual family members, we did seek to determine the importance of family structure by comparing audio recordings before and after removal of the pups at P30. The results show a clear effect of family integrity, and the sudden reduction of sonic calls following pup removal (Figure S3) could suggest that these vocalizations are produced selectively by pups.

However, there is ample evidence that adult gerbils also produce sonic vocalizations. For example, a number of low frequency call types are used by adults during a range of social interactions (Ter-Mikaelian et al., 2012; Furuyama et al., 2022), some of which are similar to a low frequency call type used by pups (Silberstein et al., 2023). Vocalization patterns of developing gerbils depend on isolation or staged interactions. Thus, when gerbil pups are recorded during isolation, ultrasonic vocalization rate declines and sonic vocalizations increase for animals that are in a high arousal state (De Ghett 1974, Silberstein et al., 2023). As gerbils progress from juvenile to adolescent development (P17-55) a significant increase in ultrasonic vocalization rate is observed during dyadic social encounters, with a distinct change in usage pattern that depends upon the sex of each animal (Holman & Seale 1991, Holman et al. 1995). The development of vocalization types has been assessed in another member of the Gerbillinae subfamily, called fat-tailed gerbils (Pachyuromys duprasi), during isolation and handling. Here, the number of ultrasonic vocalization syllable types increase from neonatal to adult animals (Zaytseva et al. 2019), while some very low frequency sonic call types were rarely observed after P20 (Zaytseva et al. 2020). By comparison, mouse syllable usage changes during development, but pups produced 10 of the 11 syllable types produced by adults (Grimsley et al. 2011). In summary, our understanding of the maturation of vocalization usage remains limited by our inability to obtain longitudinal data from individual animals within their natural social setting. For example, when recorded in their natural environment, chimpanzees display a prolonged maturation of vocalization complexity, such as the probability of a unique utterance in a sequence, with the greatest changes occuring when animals begin to experience non-kin social interactions (Bortolato et al. 2023).”

(2) Developmental progression, particularly during pre-weaning periods when pup vocal output remains unstable, might be another factor influencing cross-family vocal differences. Representing data from this non-stationary process as an overall density map could result in the loss of time-dependent information. For instance, were dominating call types consistently present throughout the recording period, or were they prominent only at specific times? Displaying the evolution of the density map would enhance understanding of this aspect.

This is a great suggestion. Thank you for bringing it up. To address this, we have added an additional figure (Figure 4) to the main text (Note that the former Figure 4 is now Figure 5). New text associated with this new figure was added to the Results and Discussion sections:

Results

“Vocal usage differences remain stable across days of development It is possible that the observed vocal usage differences could result from varying developmental progression of vocal behavior or overexpression of certain vocal types during specific periods within the recording. To assess the potential effect of daily variation on family specific vocal usage, we visualized density maps of vocal usage across days for each of the families (Figure 4A). There are two noteworthy trends: (1.) the density map remains coarsely stable across days (rows) and (2.) the maps look distinct across families on any given day (columns). This is a qualitative approximation for the repertoire’s stability, but does not take into account variation of call type usage (as defined by GMM clustering of the latent space). Figure 4B, shows the normalized usage of each cluster type over development for each family. Cluster usages during the period of “full family, shared recording days” (postnatal days beneath the purple bars) are stable across days within families – as is apparent by the horizontal striations in the plot – though each family maintains this stability through using a unique set of call types. This is addressed empirically in Figure 4C, which shows clearly separable PCA projections of the cluster usages shown in Figure 4B (purple days). Finally, we computed the pairwise Mean Max Discrepancy (MMD) between latent distributions of vocalizations from individual recording days for each of the families (Figure 4D). This shows that across-family repertoire differences are substantially larger than within-family differences. This is visualized in a multidimensional scaling projection of the MMD matrix in Figure 4E.”

Discussion

“The described family differences collapse data from multiple days into a single comparison, however it’s possible that factors such as vocal development and/or high usage of particular vocal types during specific periods of the recording could explain family differences. Therefore, we took advantage of the longitudinal nature of our dataset to assess whether repertoire differences remain stable across time. First, we visualized vocal repertoire usage across days as either UMAP probability density maps (Figure 4A) or daily GMM cluster usages (Figure 4B). Though qualitative, one can appreciate that family repertoire usage remains stable across days and appears to differ on a consistent daily basis across families. To formally quantify this, we first projected GMM cluster usages from Figure 4B into PC space and show that family GMM cluster usage patterns are highly separable, regardless of postnatal day (Figure 4C). If families had used a more overlapping set of call types, then the projections would have appeared intermixed. Next, we performed a cluster-free analysis by computing the pairwise MMD distance between VAE latent distributions of vocalizations from each family and day (Figure 4D). This analysis shows very low MMD values across days within a family (i.e. the repertoire is highly consistent with itself), and high MMD values across families/days (greater than would be expected by chance; see shuffle control in Figure S2D). The relative differences in this matrix are made clear in Figure 4E, which provides additional evidence that family vocal repertoires remain stable across days and are consistently different from other families. Taken together, we believe that this is compelling evidence that differences in vocal repertoires between families are not driven by dominating call types during specific phases in the recording period; rather, families consistently emit characteristic sets of call types across days. This opens up the possibility to assess repertoire differences over much shorter time periods (e.g. 24 hours) in future studies.”

(3) Family-specific vocalizations were credited to the transition structure, a finding that may seem obvious if the 1-gram (i.e., the proportion of call types) already differs. This result lacks depth unless it can be demonstrated that, firstly, the transition matrix provides a robust description of the data, and secondly, different families arrange the same set of syllables into unique sequences.

Thank you for these important suggestions. We agree that it is true that the 2-gram transition structure must vary based on the 1-gram structure. To determine whether this influences the interpretation of the finding, we have added Figure S5 and the following text in the Results section:

“To determine whether differences in 1-gram structure contribute to differences in the transition (2-gram) structure, we performed a number of controls. Although subtle, vertical streaks are clearly present in shuffled transition matrices that correspond to 1-gram usages (Figure S5A-B). Given the shuffled data structure, we sought to determine whether the observed transition probabilities differed significantly from chance levels. We randomly shuffled label sequences 1000 times independently for each family to generate a null transition matrix distribution. Using these null distributions and the observed transition probabilities, we computed a p-value for each transition using a one-sample t-test and created a binary transition matrix indicating which transitions happen above chance levels (Figure S5C, black pixels, p <= 0.05 after post hoc Benjamini-Hochberg multiple comparisons correction). As is made clear in Figure S5C, most transitions for each family occur significantly above chance levels, despite the inherent 1-gram structure. Moreover, by looking at transitions from a highly usage cluster type used roughly the same proportion across families (cluster 12), we show that families arrange the same sets of vocal clusters into unique sequences (Figure S5D). We believe that this provides compelling evidence that the 1-gram structure does not change the interpretation of the main claim that transition structure varies by family. “””

To address your second point, we inspected frequent transitions from individual syllables to all other syllables using bigram transition probability graphs. This revealed a common trend that across all families, many shared and unshared transitions existed, suggesting that families use the same sets of syllables to make unique transition patterns. Figure S5D shows a single syllable example of the phenomenon, with red lines indicating the shared transition types between families and black showing transition patterns not shared between families (i.e. unique family-specific transitions, or lack thereof).”

**Reviewer #2 (Public Review):**
Peterson et al., perform a series of behavioral experiments to study the repertoire and variance of Mongolian gerbil vocalizations across social groups (families). A key strength of the study is the use of a behavioral paradigm which allows for long term audio recordings under naturalistic conditions. This experimental set-up results in the identification of additional vocalization types. In combination with state of the art methods for vocalization analysis, the authors demonstrate that the distribution of sound types and the transitions between these sound types across three gerbil families is different. This is a highly compelling finding which suggests that individual families may develop distinct vocal repertoires. One potential limitation of the study lies in the cluster analysis used for identifying distinct vocalization types. The authors use a Gaussian Mixed Model (GMM) trained on variational auto Encoder derived latent representation of vocalizations to classify recorded sounds into clusters. Through the analysis the authors identify 70 distinct clusters and demonstrate a differential usage of these sound clusters across families. While the authors acknowledge the inherent challenges in cluster analysis and provide additional analyses (i.e. maximum mean discrepancy, MMD), additional analysis would increase the strength of the conclusions. In particular, analysis with different cluster sizes would be valuable. An additional limitation of the study is that due to the methodology that is used, the authors can not provide any information about the bioacoustic features that contribute to differences in sound types across families which limits interpretations about how the animals may perceive and react to these sounds in an ethologically relevant manner.The conclusions of this paper are well supported by data, but certain parts of the data analysis should be expanded and more fully explained.• Can the authors comment on the potential biological significance of the 70 sound clusters? Does each cluster represent a single sound type? How many vocal clusters can be attributed to a single individual? Similarly, can the authors comment on the intra-individual and inter-individual variability of the sound types within and across families?

Previous work documenting the Mongolian gerbil repertoire (Ter-Mikaelian 2012, Kobayasi 2012) has revealed ~12 vocalization types that vary with social context. Our thinking is that we are capturing these ~12 (plus a few more, as illustrated in Figure 2C) as well as individual or family-specific variations of some call types. Although the number of discrete call types is likely less than 70, it’s plausible that variation due to vocalizer identity pushes some calls into unique clusters. This idea is supported by the fact that both naked mole rats and Mongolian gerbils have been shown to exhibit individual-specific variation in vocalizations, though only in single call types (Barker 2021, Figure 1; Nishiyama 2011, Table I). The current study is not ideal to test this prediction, as we cannot attribute each vocalization to individual family members. Using our 4-mic array, we attempted to apply established sound source localization techniques to assign vocalizations to individuals (Neunuebel 2015), but the technique failed, presumably due to high amounts of reverberation in the arena. We are currently developing a custom deep learning based sound localization algorithm, and had hoped to extract individual animal vocalizations from our data set (part of the reason why this manuscript has taken longer than expected to return!), but the performance is not yet satisfactory for large groups of animals. We have added text to the Methods sections with the context outlined above to further justify the use of ~70 clusters.

• As a main conclusion of the paper rests on the different distribution of sound clusters across families, it is important to validate the robustness of these differences across different cluster parameters. Specifically, the authors state that "we selected 70 clusters as the most parsimonious fit". Could the authors provide more details about how this was fit? Specifically, could the authors expand upon what is meant by "prior domain knowledge about the number of vocal types...". If the authors chose a range of cluster values (i.e. 10, 30, 50, 90) does the significance of the results still hold?

Thank you for the suggestion, this is an important point that we have addressed with new analyses in the revision (see GMM clustering methods and new Figure S4). The prior domain knowledge referenced is with respect to the information known about the Mongolian gerbil vocal types provided in the response above. We have made this more clear in the discussion.

We mainly based our selection of the number of clusters using the elbow method on GMM held-out log likelihood (Figure S2C). Around 70 clusters is when the likelihood begins to plateau, though it’s clear that there are a number of reasonable cluster sizes. To assess whether cluster size has an effect on interpretation of the family differences result, we added Figure S5, where we varied the number of GMM clusters used and compared cluster usage differences across families (Figure S4A). We quantified pairwise family differences in cluster usage by computing the sum of the absolute value of differential cluster usages, for each GMM cluster value (Figure S4B). We find that relative usage differences remain unchanged across the range of cluster values used, indicating that GMM cluster size does bias the finding.

• While VAEs are powerful tools for analyzing complex datasets in this case they are restricted to analysis of spectrogram images. Have the authors identified any acoustic differences (i.e. in pitch, frequency, and other sound components) across families?

Though it’s true that this VAE is limited to spectrograms, the VAE latent space has been shown to correspond to real acoustic features such as frequency and duration, and contain a higher representational capacity than traditional acoustic features (Goffinet 2021, Figure 2). Therefore, clustering of the latent space necessarily means that vocalizations with similar acoustic features are clustered together regardless of their family identity.

Despite this, your point is well taken that there could be systematic differences in certain acoustic features for specific call types. We are not able to ascertain this with the current dataset. This is addressed in Barker 2021 by recording a single call type (soft chirp) from individuals within and across families. Mongolian gerbils have been shown to exhibit individual differences in the initial, terminal, minimum, and maximum frequency of the ultrasonic up-frequency modulated call type (Figure 2, top right green; Nishiyama 2011, Figure 1A). Therefore it’s possible that family-specific differences exist for that particular call type. To assess whether other call types show family or individual differences, it’s necessary to either (1) elicit all call types from an animal in isolation or (2) determine vocalizer identity in social-vocal interactions. The problem with the former idea is that gerbils only produce up-frequency modulated USVs in isolation and there is no known way to elicit the full vocal repertoire in single animals. The latter idea would allow for full use of the vocal repertoire, but requires invasive techniques (e.g., skull-implanted microphones, or awake-behaving laryngeal nerve recordings) that permit assignment of vocalizations to individuals during a natural social interaction. We are actively exploring solutions to both problems.

It’s likely that future studies will look deeper into acoustic differences between individuals and families. Therefore, we have added acoustic feature quantification of vocalizations in each of the GMM clusters as a reference (Figure S6).

**Reviewer #3 (Public Review):**
Summary:In this study, Peterson et al. longitudinally record and document the vocal repertoires of three Mongolian gerbil families. Using unsupervised learning techniques, they map the variability across these groups, finding that while overall statistics of, e.g., vocal emission rates and bout lengths are similar, families differed markedly in their distributions of syllable types and the transitions between these types within bouts. In addition, the large and rich data are likely to be valuable to others in the field.Strengths:- Extensive data collection across multiple days in multiple family groups.- Thoughtful application of modern analysis techniques for analyzing vocal repertoires. - Careful examination of the statistical structure of vocal behavior, with indications that these gerbils, like naked mole rats, may differ in repertoire across families.Weaknesses:- The work is largely descriptive, documenting behavior rather than testing a specific hypothesis.- The number of families (N=3) is somewhat limited.

We agree that the number of families is relatively small. However, our new analysis of vocal repertoire by postnatal day (Figure 4) demonstrates that the finding is quite robust. A high sample-size study was outside the scope of this initial observational study given the difficulty of obtaining and processing longitudinal data of this scale. In light of new analyses in Figure 4, we are confident that future studies will not need so much data to characterize family-specific differences. A single 24-hour recording should be sufficient, making comparison of many more families relatively straightforward.

**Recommendations for the authors:**

**Reviewer #1 (Recommendations For The Authors):**
Several minor concerns:(1) The three thresholds used for vocalization segmentation lack explanation.Figure 1C's first vocal event appears to define the first gap via the gray threshold (th_2, as the trace does not cross the black line) and the second gap via the black threshold (th_1 or th_3). And this is not addressed in the Methods section.

Thank you for bringing this to our attention. We agree, this is presented in an unnecessarily complicated way. We have updated the methods section describing the thresholding procedure.

“Sound onsets are detected when the amplitude exceeds 'th_3' (black dashed line, Figure 1C), and sound offset occurs when there is a subsequent local minimum e.g., amplitude less than 'th_2' (gray dashed line, Figure 1C), or 'th_1' (black dashed line, Figure 1C), whichever comes first. In this specific use case, th_2 (5) will always come before th_1 (2), therefore the gray dashed line will always be the offset. A subsequent onset will be marked if the sound amplitude crosses th_2 or th_3, whichever comes first. For example, the first sound event detected in Figure 1C shows the sound amplitude rising above the black dashed line (th_3) and marks an onset. Subsequently, the amplitude trace falls below the gray dashed line (th_2) and an offset is marked. Finally, the amplitude rises above th_2 without dipping below th_3 and an onset for a new sound event is marked. Had the amplitude dipped below th_3, a new sound event onset would be marked when the amplitude trace subsequently exceeded th_3 (e.g. between sound event 2 and 3, Figure 1C). The maximum and minimum syllable durations were selected based on published duration ranges of gerbil vocalizations (Ter-Mikaelian et al. 2012, Kobayasi & Riquimaroux, 2012).”

(2) The determination of multi-syllabic calls could be explained further. In Figure 1C, for instance, do syllables separated by short gaps (e.g., the first syllable and the rest of the first group, and the third group in this example) belong to the same call or different calls?

We have added an operational definition of mono vs. multisyllabic calls in the Results section:

“Vocalizations occur as either single syllables bounded by silence (monosyllabic) or consist of combinations of single syllables without a silent interval (multisyllabic).”

Under this definition, the examples you mentioned in Figure 1C are considered monosyllabic. One could reasonably expand the definition to include calls separated by less than X ms of silence for example, however we choose not to do that in this study. A deeper understanding of the phonation mechanisms for different gerbil vocalization types would be helpful to more rigorously determine the distinction between mono vs. multisyllabic vocalizations.

(3) Labeling the calls shown in Fig. 3D in the latent feature space would help highlight within-family diversity and between-family similarities.

Great suggestion. We have updated Figure 3 to include where in UMAP space each family’s preferred clusters are.

(4) In the introduction, the statement, "Therefore, our study considers the possibility that there is a diversity of vocalizations within the gerbil family social group" doesn't naturally follow from the previous example. This could be rephrased.

Agreed, thank you. We revised this section of the introduction to flow better.

**Reviewer #2 (Recommendations For The Authors):**
While outside the scope of the current study the authors may consider the following experiments and analysis for future studies:• Do vocal repertories retain their family signatures across subsequent generations of pups? (i.e. if vocalizations are continually monitored during second or third litters of the same parents).• Do the authors observe any long-term changes in family repertoires related to the developmental trajectory of the pups? Are there changes in individual pup vocal features or sound type usage throughout development?

Thank you for these great suggestions. Given that naked mole rats learn vocalizations through cultural transmission, it would be interesting to see whether other subterranean species with complex social structures (gerbils, voles, rats) have similar abilities. A straightforward way to assess this possibility could be as you suggest — are latent distributions of vocalizations from multi-generational families closer together than cross-family differences? If true, this would provide compelling evidence to investigate further.

We partially address your second suggestion in our response to Reviewer 1 and in Figure S4, which shows that the family repertoire remains stable throughout this particular period of development. This doesn’t rule out the possibility that there could be other phases of development that undergo more vocal change. Your final suggestion is an area that we are actively researching and eager to know the answer to. A follow-up question: could differences in pup vocal features contribute to differential care by parents?

**Reviewer #3 (Recommendations For The Authors):**
In all, I found the paper clearly written and the figures easy to follow. One small suggestion:Figure 1: I can't see the black and gray thresholds described in the caption very well. Perhaps a zoom-in to the first 0.15s or so of the normalized amplitude plot would better display these.

Agreed, thank you. We added a zoom-in to Figure 1.